# Hyperglycemia and Hyperlipidemia with Kidney or Liver Transplantation: A Review

**DOI:** 10.3390/biology12091185

**Published:** 2023-08-29

**Authors:** John A. D’Elia, Larry A. Weinrauch

**Affiliations:** Kidney and Hypertension Section, E P Joslin Research Laboratory, Joslin Diabetes Center, Department of Medicine, Beth Israel Deaconess Medical Center, Harvard Medical School, Boston, MA 02215, USA; jd'elia@joslin.harvard.edu

**Keywords:** solid organ allografts: liver, kidney, hyperglycemia, diabetes, hyperlipidemia, allograft survival, transplantation, calcineurin Inhibitors, mTOR inhibitors

## Abstract

**Simple Summary:**

Post-allograft transplant antirejection regimens (glucocorticoids, azathioprine, mycophenolate, calcineurin inhibitors and mTOR inhibitors) may trigger or aggravate hyperglycemia or hyperlipidemia. Post-transplant medication management must balance immune suppression and glucose and lipid control because the existence of glucose or lipid imbalance is associated with shorter times of useful allograft function. We review the underlying mechanism of relationships between glycemia/lipidemia control, transplant rejection and graft aging.

**Abstract:**

Although solid organ transplantation in persons with diabetes mellitus is often associated with hyperglycemia, the risk of hyperlipidemia in all organ transplant recipients is often underestimated. The diagnosis of diabetes often predates transplantation; however, in a moderate percentage of allograft recipients, perioperative hyperglycemia occurs triggered by antirejection regimens. Post-transplant prescription of glucocorticoids, calcineurin inhibitors and mTOR inhibitors are associated with increased lipid concentrations. The existence of diabetes mellitus prior to or following a liver transplant is associated with shorter times of useful allograft function. A cycle involving Smad, TGF beta, m-TOR and toll-like receptors has been identified in the contribution of rejection and aging of allografts. Glucocorticoids (prednisone) and calcineurin inhibitors (cyclosporine and tacrolimus) induce hyperglycemia associated with insulin resistance. Azathioprine, mycophenolate and prednisone are associated with lipogenesis. mTOR inhibitors (rapamycin) are used to decrease doses of atherogenic agents used for immunosuppression. Post-transplant medication management must balance immune suppression and glucose and lipid control. Concerns regarding rejection often override those relative to systemic and organ vascular aging and survival. This review focuses attention on the underlying mechanism of relationships between glycemia/lipidemia control, transplant rejection and graft aging.

## 1. Introduction

Despite diabetes mellitus being the most prevalent underlying condition requiring transplantation, all solid organ recipients are subject to hyperlipidemic vascular complications. Underlying mechanisms explaining vascular risk associated with immunosuppressive antirejection medication have not been widely disseminated.

Type 2 diabetes mellitus, predominantly a state of insulin resistance with increased lipogenesis, is initially associated with hyperinsulinemia that may decrease with time. We now recognize subtypes of type 2 diabetes that may be characterized by remission: gestational remitting with delivery, obesity-related remitting with bariatric surgery and steroid or immunosuppression related remitting with therapy cessation. In some patients, the appearance of hyperglycemia exposes a trait that may lead to the well-known complications of diabetes (nephropathy, neuropathy, retinopathy and early vascular disease). However, in others, prolonged follow-up does not result in these complications. The prognostic impact of these subtypes within the transplant population is unknown.

In the 50 years of genetically non-related kidney transplantation, there was a gradual acceptance of recipients with diabetes mellitus, followed by awareness that hyperglycemia and infection were important side effects of immunosuppression [1]. The US Renal Data System report for 2000–2018 included 70,000 persons with diabetes mellitus who had received a kidney transplant [2]. The survivals for 50,000 persons with type 2 diabetes were 94% at 1 year, 77% at 5 years, 48% at 10 years and 25% at 15 years [2].

Individuals with type 1 diabetes mellitus were younger, demonstrating higher survivals at these time intervals (96%, 85%, 65% and 46%). Among over 10,000 persons with native kidneys, diabetes and hypertension, modulation of medication dosages for control of hyperglycemia was noted to be related to the estimated glomerular filtration rate [3]. Modulations of immunosuppressive regimens were also observed in the FAVORIT study involving 1500 kidney transplant recipients with diabetes mellitus morbidity [4]. Demonstration that infection and malignancy were the most frequent causes of post-transplant mortality among diabetic transplant recipients prompted the quest to eliminate causes of hyperglycemia [5]. Whether acquired diabetes mellitus is found before or after kidney transplantation, the mechanism of vascular complications relates to extracellular glycation of protein and lipid components of cell structures with cross-linking that alters function [6,7]. Several immunosuppressive agents (prednisone, calcineurin inhibitors and mTor antagonists) contribute to post-transplant hyperglycemia. Recently appreciated intracellular alterations involving protein kinases (especially series of protein kinase C) have become the focus of attention.

## 2. Mammalian Target of Rapamycin in Diabetes Mellitus: Nephropathy and Kidney Transplantation

Mammalian target of rapamycin (mTOR) is a complex of protein kinase signal molecules that promote kidney disease. Located in the glomerular podocyte and mesangial cell, mTOR is involved in the generation of diabetic nephropathy (Table 1). The mTOR complex located in the renal tubular epithelial cell is involved in the generation of polycystic kidney disease [8,9]. The mTOR inhibitor sirolimus (rapamycin) was discovered 50 years ago. Sirolimus has immunosuppressive effects as well as anti-tumor and anti-fungal effects [10]. Two viruses best known for complicating the course of kidney allograft recipients are cytomegalovirus (CMV) and BK virus. Both viruses can injure allograft function. Kidney allograft host cells provide activated mTOR for CMV and BKV replication. Several studies suggest that anti-mTOR immunosuppressive agents have antiviral benefits as opposed to other immunosuppressive medications [11,12,13]. Following receipt of renal allografts, the prevalence of squamous cell skin cancer is lower with mTOR inhibitors than with other immunosuppressive agents [10]. Because combinations of immunosuppressive agents may be required, calcineurin inhibitors (such as cyclosporine), which can be nephrotoxic and diabetogenic at high doses, may be combined with mTOR Inhibitors. Some investigations recommend a dose of cyclosporine low enough to avoid nephrotoxicity to the allograft along with a dose of rapamycin low enough to avoid lipotoxicity to the heart [14,15,16].

## 3. Hyperglycemia Associated with Medications for Transplant Immunosuppression

Three excellent articles introduce the topic of medications for immunosuppression with the timeline of post-transplantation hyperglycemia [17,18,19]. To the best of our knowledge, azathioprine and mycophenolate do not generally cause hyperglycemia or hyperlipidemia. Transplant allograft recipients treated with azathioprine may have increased concentrations of triglycerides (Table 1) [20]. Mycophenolate mofetil treatment reduces the risk of plaque formation in the atherosclerosis-prone rabbit on a high-cholesterol diet [21].

The medication most often associated with hyperglycemia or the concept of “new onset diabetes after transplantation” in the years after kidney transplantation is prednisone [22]. Mechanisms for insulin resistance and allograft injury with glucocorticoids include steroids that make the liver less sensitive to insulin, so it carries on releasing glucose even if the pancreas is releasing insulin [23]; steroids that cause increased appetite, fluid retention and weight gain; and increased gluconeogenesis—the conversion of muscle protein amino acids into glucose stored in the liver as glycogen. This side effect is not as potent as lipogenesis in the problem of insulin resistance (Table 2). In addition, there is increased lipogenesis, the conversion of muscle protein amino acids into fatty acids stored in visceral adipose tissue and the liver as triglycerides. Osteocalcin, secreted by osteoblasts, functions to promote bone strength. Another function of osteocalcin is the suppression of lipogenesis in the liver and visceral adipose tissue. Therefore, the use of glucocorticoids, which inhibit the action of osteocalcin, also results in lipogenesis with resultant insulin resistance [24,25]. Finally, toxic hyperglycemia in human [26] and experimental animal [27] renal tubular epithelial cells involve caspase enzymes that generate hydrogen peroxide.

As a result of the mechanisms listed in Table 2, tacrolimus (and cyclosporine, both calcineurin inhibitors) decreases insulin secretion by the suppression of expression of genes for beta cell function, resulting in a decreased level of insulin mRNA [28] and inhibition of the activity of glucokinase, thereby decreasing Beta cell response to increases in glucose concentration [29] and reducing the number of pancreatic Beta cells by inducing apoptosis of Beta cells [30].

Rapamycin and other mTOR inhibitors eliminate the insulin resistance associated with protein kinase-related hypertriglyceridemia [31]. This mechanism has been studied extensively in the streptozotocin (STZ) diabetic rat (type 1 diabetes model). In this model, kidney glomerular mesangial cells exposed to increased levels of glucose were associated with increased activity of Protein Kinase C (PKC) beta isoform [32]. Activated PKC beta isoforms were associated with increased expression of Transforming Growth Factor (TGF) Beta. TGF Beta is known to be associated with increased production of fibronectin and type 4 collagen (Table 3), which leads to mesangial expansion and basement membrane thickening in glomerular cells in diabetes mellitus. Pericytes attached to capillaries in the retina are potentially susceptible to hyperglycemia associated with advanced glycated end-products, which cross-link pathologically [33]. Additional studies found pathological changes in the vascular layer of the retina [34]. The progression of microvascular changes (loss of perivascular support cells and capillary aneurysms) has been documented in both the STZ diabetic rat and the transgenic mouse. Inhibitors of PKC isoform beta have been shown to prevent pathological findings in the renal glomerulus and ophthalmic retina [35,36,37].

The Banff Human Organ Transplantation study identified the peritubular portion of the capillary as the target for antibody attack in chronic active transplant rejection of kidney allografts [38,39]. The process is referred to as capillaritis (Table 4). The well-known cardiac complication of hyperglycemia, cardiomyopathy, considered to be the result of the cross-linking of advanced glycated end-products, causes stiffness in relaxation and contraction [40]. A gene factor for remodeling in diabetic cardiomyopathy, known as Suppressor of mothers against decapentaplegic (Smad-3), operates in cooperation with TGF-Beta [41]. Weakness in retinal capillaries leading to aneurysms may be due to the inflexibility of retinal pericytes. Thus, capillaries in kidney allografts now shown to be the target of antibodies in chronic active rejection may also be the subject of weakening through cross-linking of advanced glycated end-products (Table 4).

Investigators have described a cascade: Smad → TGF Beta → mTOR → mesangial collagen fibrosis inhibited by rapamycin [42]. This rapamycin effect could be over-ridden by hypoxia-inducible factor (HIF) 1 alpha at normal oxygen levels.

Further investigation into the role of the important member of mTOR, the protein kinase PKC, was centered upon early diabetic nephropathy (albuminuria) in the STZ diabetic rat. At this early stage, innate immunity (toll-like receptors) was activated along with cytokines of inflammation (interleukin 17). Inhibition of Protein Kinase C by rapamycin was associated with the blockade of progression of nephropathy in this experimental model of Type 1 diabetes mellitus [35]. There is an mTOR tuberin pathway. Tuberin inhibits cell cycle progression and, therefore, has a negative impact on the life cycle of renal tubular cells. Hyperglycemia contributes to the apoptosis of renal tubular epithelial cells under the influence of tuberin and is blocked by insulin [36]. In the mTOR/tuberin pathway, cell size regulation likely occurs by activation/inhibition of tuberin vs. inhibition/activation of mTOR [36]. Thus, the insulin Akt cascade both activates and downgrades tuberin with opposing effects on MTOR, thereby regulating cell size.

## 4. Lipogenesis before/after Transplantation

### 4.1. Kidney Allografts

Pericardiac and perirenal fat are predictors of atherosclerotic pathology affecting cardiovascular outcomes and renal function. When perirenal fat volume was studied, live donor renal grafts with higher fat volume resulted in lower estimated GFR than those with lesser perirenal fat volume [43].

Cholesterol is essential for the structure of the glomerular podocyte diaphragm, which functions to select molecules for excretion versus those to be retained. Excess levels of ceramide, a sphingolipid, can be destructive to the ability to discriminate solutes critical for excretion from those appropriate for retention [44]. Increased concentrations of oxidized low-density lipoproteins are toxic to podocytes of glomeruli in membranous and diabetic nephropathy, both of which may recur allograft during follow-up [45]. Excess levels of triglyceride may inhibit tubular resorption of electrolytes and amino acids [46]. A deficiency of peroxisome proliferator-activated receptor (PPAR) gamma has been found to have increased expression of nuclear factor kappa beta associated with inflammation/fibrosis [47]. Among experimental animals exposed to thiazolidinediones, synthetic PPAR gamma agonists, kidney function was preserved [47].

It stands to reason that some of these pre-transplantation mechanisms will be operating after transplantation, depending upon the choice of immunosuppression [48]. The provision of pancreas allograft in addition to kidney allograft appears to produce better cardiovascular outcomes than kidney transplant alone among diabetic recipients [48,49,50,51,52]. Differences in endothelial-dependent dilation of the brachial artery and intimal thickness of the carotid artery [48,49] may contribute to the differences observed in long-term follow-up. Hyperlipidemia experimental models are available for studies of azathioprine and mycophenolate [20,21] as noted in Table 1.

### 4.2. Liver Allografts

Although liver insufficiency does not commonly occur after a kidney transplant, renal insufficiency may occur after a liver transplant. Among twelve liver allograft recipients studied before and after transplant, an observed fall in the glomerular filtration rate (GFR (27%) was not due to dehydration or heart failure as evidenced by increases in both somatic blood flow (RBF) and oxygen utilization following transplantation [53].

Follow-up of recipients of liver allografts indicates significantly better results for those without a diagnosis of diabetes mellitus compared to those with diabetes mellitus [54]. Pretransplant adult-onset diabetes occurring as liver function decreases to a state of fibrosis known as cirrhosis and has been termed “hepatogenous” diabetes [55]. Glucose tolerance testing pre- and post-transplant has demonstrated a new type 2 diabetes post-transplant [56]. Hepatic lipotoxicity is associated with cholesterol, ceramides, sphingolipids and triglycerides. These lipid molecules may be encapsulated into droplets as intracellular organelles that may cause mitochondrial injury and stress the endoplasmic reticulum [57,58,59].

Since some of the mechanisms of lipogenesis are associated with excess nutrition or elevated glucocorticoid dose, there is a chance for recurrence of the process after transplantation. Excessive caloric intake or toxic ingestion results in steatosis/fibrosis of the allograft. There is no evidence for allograft-related lipogenesis [60].

Glucocorticoids contribute to lipogenesis by the conversion of skeletal muscle amino acids through acetate to glycogen via glycogen synthase. Similarly, glucocorticoids organize the conversion of skeletal muscle amino acid through acetate to palmitate with the stimulation of fatty acid synthase [61,62,63]. Nonalcoholic fatty liver disease is an important cause of liver failure. Since this metabolic condition is associated with insulin resistance, it will recur in liver transplant allografts, especially if glucocorticoid immunosuppressive doses are high [64,65]. The calcineurin inhibitor, cyclosporin, was associated with elevated levels of serum cholesterol, which were reduced when another calcineurin inhibitor, tacrolimus, was introduced into the transplantation immunosuppressive program [66].

## 5. Management of Post-Transplant Hyperglycemia

Minimal to moderate hyperglycemia following solid-organ transplantation may not require treatment and may not ordinarily be called post-transplant diabetes mellitus [67,68]. A key report indicated that the post-kidney or post-liver-transplantation new onset of diabetes mellitus may be as high as 20% in each instance [29]. Three publications involving the term post-transplantation diabetes mellitus pinpoint calcineurin inhibitors (tacrolimus and cyclosporine) as fundamental since glucocorticoid doses would have been significantly diminished [19,69,70,71], establishing the definition of post-transplantation diabetes mellitus as continued hyperglycemia beyond 45 days (6+ weeks) of treatment, emphasizing the key mechanism of toxicity of cyclosporine to be the removal of the cell surface glucose transporter. A Lewis rat model was described, quantifying the recurrence of coronary artery atherosclerosis in heart transplant allografts [71]. In this study, both groups received cyclosporine and a high-fructose diet to induce post-heart transplant hyperlipidemia. However, only the group rendered insulin-dependent by the use of streptozotocin developed severe coronary artery atherosclerosis in short-term follow-up [71].

When kidney allograft function falls below an estimated glomerular filtration of 60 mL/minute (Table 5), there is a need to reduce doses of certain sulfonylurea medications (glyburide), dipeptidyl dipeptidase receptor 4 inhibitors (sitagliptin), biguanides (metformin) and insulin since these medications are renally excreted. The sulfonyl urea glipizide is cleared by liver function. Since sodium/glucose co-transporter 2 inhibitor (canagliflozin, empagliflozin and dapagliflozin) efficiency is reduced at this lower end of the estimated glomerular filtration rate, prescription is not recommended. On the other hand, there is active consideration for the use of glucagon-like peptide 1 receptor agonists (exenatide, liraglutide, dulaglutide) in the setting of calcineurin inhibition or tacrolimus-related hyperglycemia [72,73].

Since the estimated glomerular filtration rate (eGFR) may change rapidly with unrelated kidney allografts, doses of antihyperglycemic medications may require rapid change. Insulin is metabolized and excreted by the kidney, and levels rise with deceasing allograft function. C-Peptide, also renally excreted, is a more reliable measure of circulating insulin that is removed in part with hepatic passage through the liver. The glucose/C-Peptide ratio is useful in decisions of doses of medications for glucose control, especially when eGFR is changing [74]. The normal level for glucose/C-Peptide is 55 ± 25 mg/dL/ng/mL (range of 30–80). A high glucose/C-Peptide concentration (>120) is consistent with the need to initiate insulin injections in addition to the other classes being used. A low glucose/C-Peptide ratio (<20) is consistent with hypoglycemia unless all anti-hyperglycemic medication doses are lowered.

In rare instances, all doses of medications for glucose control are discontinued as eGFR is falling [75], leading to the return to dialysis dependency for the kidney transplant recipient.

## 6. Management of Post-Transplant Hyperlipidemia

Post-transplant prescription of glucocorticoids (prednisone and dexamethasone), calcineurin inhibitors (cyclosporin and tacrolimus) and mTOR inhibitors (sirolimus and everolimus) is associated with increased lipid concentrations. Glucocorticoids increase triglycerides through the inhibition of lipoprotein lipase, calcineurin inhibitors also inhibitors lipoprotein lipase [76]. Lower levels of post-transplantation lipids have been identified with belatacept. The mechanism is associated with a pathway involving the co-stimulation of antigen-bearing B lymphocytes and antibody-producing T lymphocytes [76].

In addition to the well-known adverse cardiovascular outcomes of long-term hyperlipidemia and their reduction with lipid-lowering agents, there is now significant research linking elevated lipids to allograft rejection and vasculopathy [76,77,78,79,80,81,82,83]. Mechanisms for allograft vasculopathy include the activation of smooth muscle endothelium [77,78,79]. The presence of hyperlipidemia is associated with excess vascular plaque formation. The reduction in serum lipid levels among transplant recipients has been demonstrated to decrease accelerated allograft vasculopathy [84]. Pharmacologic lipid reduction achieved by the use of HMG CoA reductase inhibitors may reduce antibody-mediated rejection by the inhibition of T cell proliferation and T cell interaction with antigen-bearing B lymphocytes [85]. Apolipoprotein E (Apo E) mediates the uptake of cholesterol through an ldl receptor pathway. Deletion of apo E results in a significant increase in plasma cholesterol due to failure to clear the lipoprotein. Apo E is made in macrophages, not T or B cells. In the mouse model, the deletion of apo E is associated with impaired uptake of cholesterol in the liver and elevated plasma cholesterol when compared to wild-type mice. When both types are exposed to high-fat diets, the apoE-deficient mice have higher cholesterol, and rejection of cardiac allograft occurs more rapidly. Exposure of the wild type to a high-fat diet with a slower rise in plasma cholesterol is associated with similar but slower evidence for rejection [86]. Cardiac allografts are rejected equally in wild-type vs. apo E-deficient mice, confirming the human observation of increased rejection risk associated with hyperlipidemia.

Hyperlipidemia following solid organ transplant is a consequence of the interaction of immunosuppressive agents, genetic predisposition and diet/obesity [87]. Each group of immunosuppressive agents that have been used in solid organ recipients (corticosteroids, azathioprine, mycophenolate mofetil, calcineurin inhibitors and mTor antagonists) either alone or in combination has been demonstrated to potentiate dyslipidemia and adversely affect long-term allograft outcomes.

Despite significant animal model and human research linking adverse cardiovascular outcomes to hyperlipidemia, there are no allograft-recipient-specific guidelines for the control of hyperlipidemia.

Generally, kidney and liver transplant candidates are chosen because their risk of cardiovascular events is low. For this reason, trials of preventive therapies often do not have sufficient cardiovascular events to produce meaningful statistical differences in benefits [5]. For patients in whom the allograft is cardiac, however, the need for lowering of lipid and plaque burden is more evident. Based upon the current literature and our observation, we feel that primary prevention through lipid-lowering therapy is indicated when prolonged survival should be anticipated and hyperlipidemia, hyperglycemia or frank diabetes is noted. There is also support for lipid-lowering therapy in all heart transplant recipients or when allograft function is threatened. Given the multiple medications that must be taken by organ recipients, careful attention must be paid to possible drug–drug interactions and the fact that research is relatively limited to HMG CoA reductase inhibitors (statins). For this reason, we would favor outcome studies that focus on the best serum lipid levels for transplant recipients for the preservation of allograft function and avoidance of cardiovascular events. With respect to the choice of lipid-lowering agents, one must be aware of drug–drug interactions with various immunomodulators (especially cyclosporine) and the relative lack of information demonstrating comparative benefits of newer agents when compared to the most studied HMG Co-A reductase inhibitors (statins). With respect to newer agents such as ezetimibe, PCSK9 inhibitors and small interfering RNA-based therapeutics, there are few outcome trials in organ transplant recipients to support their widespread use.

## 7. Conclusions

The presence of diabetes or hyperlipidemia of either long duration or accompanying the requirement for organ replacement therapy is associated with elevated risk to allograft function [86,87]. Glycemic control is more difficult with the use of prednisone and tacrolimus [88]. The control of hyperlipidemia is more difficult with the use of azathioprine and high-dose sirolimus. Elevated lipid levels contribute to insulin resistance. Elevated glucose levels lead to advanced glycated end-products, which may cross-link, causing stiffness, i.e., resistance to movement. The restriction of relaxation and contraction (cardiomyopathy) is the best-known result of stiffness associated with advanced glycated end-products. This process may involve pericytes around capillaries as well. Pericyte stiffness would leave the capillary at risk for the formation of micro-aneurysms, which are seen in diabetic retinopathy.

Increased risk for kidney allograft rejection is partially due to increased activity of the inflammation cascade. Activation of the pathway known as mammalian target of rapamycin occurs in persons with diabetes mellitus. This leads to increased inflammation, which contributes to allograft rejection. The Banff Transplantation Study has identified the capillary pericyte as the most sensitive site for antibody attack in chronic, active allograft rejection. We can conclude that a diagnosis of diabetes mellitus, before or after the initiation of immunosuppression, contributes to the risk of loss of allograft function, with the capillary pericyte as a major focus of pathogenesis. A cascade combining several factors in kidney fibrosis included sMAD → TGF Beta → mTOR → mesangial collagen. Rapamycin was protective by blocking the process of fibrosis, but the rapamycin effect could be over-ridden by the over-expression of HIF (Figure 1).

Beyond the scope of this review are gestational diabetes [89], obesity treated with bariatric surgery [90,91] and the treatment of renal allograft recipients with mTOR inhibition who received the SARS-CoV-2 mRNA BNT 162 bz (Pfizer BioNTech) vaccine. Their antibody response was greater than that of matched controls who did not receive mTOR inhibitors [92]. Also, beyond the scope of this review is consensus on the definition of remission of type 2 diabetes mellitus [93,94], which may be useful as transplant recipients are treated with a variety of immunosuppressive agents. We also have little information to clarify the relation between immunosuppression and observed combinations of diabetic nephropathy and membranous glomerulopathy sometimes found in renal allografts [95,96,97].

## Figures and Tables

**Figure 1 biology-12-01185-f001:**
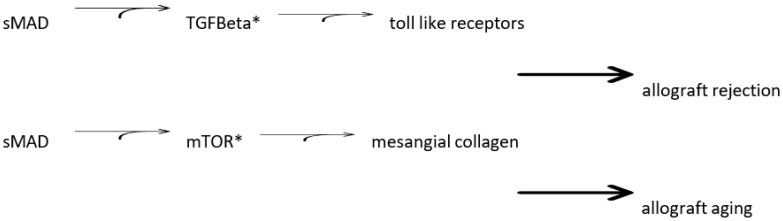
Mechanisms of allograft rejection and aging in diabetes mellitus. * inhibited by rapamycin. AGE (Advanced Glycated End-Products). sMAD (Suppressor of Mothers against Decapentaplegic Transcription Factor Family). TGF Beta (Transforming Growth Factor Beta). mTOR (mammalian Target Of Rapamycin).

**Table 1 biology-12-01185-t001:** Metabolic Effects of Post-Organ-Transplant Immunosuppression.

Drug	Dysmetabolic Effect	Observation
Prednisone	Increased lipogenesis and gluconeogenesis	Rat liver study (rat model)
Azathioprine	Increased triglycerides	Transplant patients treated with azathioprine
Mycophenolate mofetil	Decreased atheroma size	Atheroma prone rabbit on atherogenic diet
Calcineurin inhibitors (cyclosporin, tacrolimus)	Decreased insulin secretion	Humans post-organ-transplant
mTOR * inhibitors (sirolimus, temsirolimus, everolimus)	Increased insulin resistance	Hepatocytes (rat), muscle cell cultures

* mTOR (mammalian target of Rapamycin).

**Table 2 biology-12-01185-t002:** Pathogenesis of Complications of Diabetes Mellitus Associated with Protein Kinases.

Protein Kinase C: Increased levels associated with TGF Beta Increased fibronectin, type 4 collagen with complications of diabetes mellitus. Retina of eye: capillary aneurisms with bleeding. Glomerulus of kidney: fibrosis of basement membranes, mesangial cells. Increased innate immunity with rejection of kidney allografts. Toll-like receptors. Cytokines (interleukin 17). Inhibition of Na+/K+ ATPase Limits control of hypokalemia with hyper-aldosterone and hyperkalemia with ACE inhibitor. mTOR *: increased levels for recipients of kidney transplant allografts. Decreased insulin secretion: reversed by rapamycin. Decreased production of antibodies after COVID vaccine: reversed by rapamycin.

* mTOR (mammalian target of Rapamycin).

**Table 3 biology-12-01185-t003:** Contribution of mTOR * to Pathology: Protection with Rapamycin.

Kidney Glomerulus Podocyte dysfunction. Mesangial expansion. Basement membrane thickening. Diabetic nephropathy. Tubule Expression of familial polycystic disorder. Injury with BK and CM ** viruses. Post-transplantation injury with cyclosporine. Skin Post-transplantation squamous cell cancer.

* mammalian target of rapamycin. ** BK and cytomegalovirus.

**Table 4 biology-12-01185-t004:** Pathogenesis of Vascular Complications for recipients of Kidney Transplants: Pre vs. Post-Transplant Diabetes Mellitus.

Pathologic Anatomy					
	Mechanism				
		AGE *	sMAD **	TGF-Beta ***	mTOR ****
Pericyte/Fibrosis	Hyperglycemia	+	+	-	-
-	Fibronectin	+	+	-	-
	Collagen 4	+	+	-	-
Artery/Atheroma					
	Hyperlipidemia	-	-	+	+
	Toll-like receptors	-	-	+	+
	Interleukin 17	-	-	+	+
Cardiac Muscle Hypertrophy					
	Hypertension	-	-	-	-
	Fibrosis	+	+	-	-

* AGE Advanced Glycated End-Products. ** sMAD Suppressor of Mothers against Decapentaplegic Transcription Factor Family. *** TGF Transforming Growth Factor Beta. **** mTOR mammalian target of Rapamycin.

**Table 5 biology-12-01185-t005:** Medications to Control Hyperglycemia: Pre- or post-Transplantation.

GFR > 60 mL/minute: usual doses of Insulin. Metformin. Sulfonyl urea glyburide (glipizide). Dipeptidyl peptidase receptor inhibitor (sitagliptin). Glucagon-like receptor agonist (liraglutide, dulaglutide). Sodium/glucose transport inhibitor 2 (canagliflozin, empagliflozin, dapagliflozin). GFR 30–60 mL/minute: Reduced doses of Insulin. Metformin. Glyburide but not glipizide. Sitagliptin. Liraglutide and Dulaglutide. Frequently not used not used: canagliflozin, empagliflozin and dapagliflozin. GFR < 30 mL/minute: Restricted doses of Insulin. Metformin. Glyburide but not glipizide. Sitagliptin.

## Data Availability

Not applicable.

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
