# Peer review of "Hyperglycemia and Hyperlipidemia with Kidney or Liver Transplantation: A Review"

_biology, 2023, doi:10.3390/biology12091185_

Round 1

Reviewer 1 Report (Previous Reviewer 1)

The review is significantly improved, and I recommend accepting it.

Reviewer 2 Report (Previous Reviewer 2)

I think it is much better now, it has improved in quality. Good job

This manuscript is a resubmission of an earlier submission. The following is a list of the peer review reports and author responses from that submission.

Round 1

Reviewer 1 Report

I congratulate the authors for the well-prepared review. My specific comments are below:

  1. The abstract should be revised, and the aim of the review should be included.
  2. The first paragraph in the introduction (lines 19 to 32) is missing the references, which make no sense.
  3. Also, on lines 37 and 38, the authors should provide the references to this information.
  4. Minor correction in lines 42, 44, 88, 123, 222, 231, 235, 239, 304, 306, 313, and 315.
  5. Authors should use the same style of writing in the table 1 heading and content. Also the same in Table 4.
  6. Lines 88 to 100 should be written as a paragraph instead of point by point; the same is true from line 102 to line 123.
  7. Authors should draw tables 3, 5, and 6 instead of randomly putting the information point by point.
  8. Lines 207 and 208, the authors should provide the references to these information's.
  9. Lines 255–259, the authors should provide the references to these information's.
  10. Authors should use the abbreviation and avoid repeating the same words in the entire manuscript.
  11. The review should be carefully revised as there are many grammar mistakes.

Reviewer 2 Report

Thank you for the opportunity to comment on this document. But I have serious doubts that the topic deserves to be published.

1º I would like to know what is the novelty of the article and its contribution to the knowledge of the scientific community, since as we know the metabolic syndrome and hyperglycemia is something common in transplanted patients, it is not new.

2º The introduction should be expanded to provide a solid background of the main idea of the study. In addition, the authors should end the introduction with a clear hypothesis and specifying the objective of their work.

3º Methodologically speaking, the review has serious methodological problems that need to be revised, since I did not know it was a review until I saw the conclusions.